# Can a Quantum Walk Tell Which Is Which?A Study of Quantum Walk-Based Graph Similarity

**DOI:** 10.3390/e21030328

**Published:** 2019-03-26

**Authors:** Giorgia Minello, Luca Rossi, Andrea Torsello

**Affiliations:** 1Dipartimento di Scienze Ambientali, Informatica e Statistica, Universita Ca’ Foscari Venezia, via Torino 155, 30170 Venezia Mestre, Italy; 2Department of Computer Science and Engineering, Southern University of Science and Technology, Nanshan District, Shenzhen 518055, China

**Keywords:** quantum walks, graph similarity, graph kernels, directed graphs

## Abstract

We consider the problem of measuring the similarity between two graphs using continuous-time quantum walks and comparing their time-evolution by means of the quantum Jensen-Shannon divergence. Contrary to previous works that focused solely on undirected graphs, here we consider the case of both directed and undirected graphs. We also consider the use of alternative Hamiltonians as well as the possibility of integrating additional node-level topological information into the proposed framework. We set up a graph classification task and we provide empirical evidence that: (1) our similarity measure can effectively incorporate the edge directionality information, leading to a significant improvement in classification accuracy; (2) the choice of the quantum walk Hamiltonian does not have a significant effect on the classification accuracy; (3) the addition of node-level topological information improves the classification accuracy in some but not all cases. We also theoretically prove that under certain constraints, the proposed similarity measure is positive definite and thus a valid kernel measure. Finally, we describe a fully quantum procedure to compute the kernel.

## 1. Introduction

In recent years, we have observed rapid advancements in the fields of machine learning and quantum computation. An increasing number of researchers is looking at challenges emerging at the intersection of these two fields, from quantum annealing as an alternative to classical simulated annealing [1,2] to quantum parallelism as a source of algorithmic speedup [3,4]. Quantum walks [5,6], the quantum mechanical analogue of classical random walks, have been shown to provide the potential for exponential speedups over classical computation, in part thanks to an array of exotic properties not exhibited by their classic counterpart, such as interference. The rising popularity of quantum walks can be further understood by looking at the work of Childs [7], who showed that any quantum computation can be efficiently simulated by a quantum walk on a sparse and unweighted graph, thus elevating quantum walks to the status of universal computational primitives.

In machine learning and pattern recognition, graphs are used as convenient representations for systems that are best described in terms of their structure and classical random walks have been repeatedly and successfully used to analyse such structure [8,9,10,11,12]. Indeed, the rich expressiveness of these representations comes with several issues when applying standard machine learning and pattern recognition techniques to them. In fact, these techniques usually require graphs to be mapped to corresponding vectorial representations, which in turn needs a canonical node order to be established first. Graphs with different number of nodes and edges present yet another challenge, as the dimension of the embedding space depends on this information. In this context, classical random walks provide an effective way to compute graph invariants that can be use to characterise their structure and embed them into a vectorial space [13,14].

Classical random walks have also been used to successfully define graph kernels [9,12,15,16,17,18]. Graph kernels offer an elegant way to transform the problem from that of finding an explicit permutation invariant embedding to that of defining a positive semi-definite pairwise kernel measure. One can then choose between an array of different kernel methods, the best known example being support vector machines (SVMs) [19], to solve the pattern analysis task at hand. This is based on the well-known kernel trick. Given a set *X* and a positive semi-definite kernel k:X×X→R, there exists a map ϕ:X→H into a Hilbert space *H*, such that $k\left(x,y\right)=\phi {\left(x\right)}^{⊤}\phi \left(y\right)$ for all x,y∈X. Thus, any algorithm that can be formulated in terms of scalar products of the ϕ(x)’s can be applied to a set of data (e.g., vectors, graphs) on which a kernel is defined. In the case of graphs, examples of kernels include the shortest-path kernel [15], the graphlet kernel [20], and the Weisfeiler-Lehman subtree kernel [16]. The common principle connecting these kernels is that of measuring the similarity between two graphs in terms of the similarity between simpler substructures (e.g., paths, subgraphs, subtrees) contained in the original graphs. When the substructures used to decompose the graphs are classical random walks, we obtain kernels like the random walk kernel [9] or the Jensen-Shannon kernel [12]. The kernel of Gärtner et al. [9] counts the number of matching random walks between two graphs while using a decay factor to downweigh the contribution of long walks to the kernel. The kernel of Bai and Hancock [12], on the other hand, is based on the idea of using classical random walks to associate a probability distribution to each graph and then use the Jensen-Shannon divergence [21,22] between these distributions as a proxy for the similarity between the original graphs.

The quantum analogue of the Jensen-Shannon divergence (QJSD) [23] allows to extend the Jensen-Shannon divergence kernel [12] to the quantum realm [17,18]. While the classical Jensen-Shannon divergence is a pairwise measure on probability distributions, the QJSD is defined on quantum states. As its classical counterparts, the QJSD is symmetric, bounded, always defined, and it has been proved to be a metric for the special case of pure states [23,24]. Unfortunately, there is no theoretical proof yet that the same holds for mixed states. Inspired by this quantum divergence measure, Rossi et al. [18] and Bai et al. [17] have proposed two different graph kernels based on continuous-time quantum walks. While the kernel of Bai et al. was proved to be positive semi-definite [17], it has several drawbacks compared to [18] (for which the positive semi-definiteness has not been proved yet), most notably the need to compute the optimal alignment between the input graphs before the quantum walk-based analysis can commence. Rossi et al. [18], on the other hand, are able to avoid this by exploiting the presence of interference effects. Given a pair of input graphs, their method establishes a complete set of connections between them and defines the initial states of two walks on this structure so as to highlight the presence of structural symmetries [18,25]. However the analysis of Rossi et al. [18] is limited to the case of undirected graphs and, as mentioned, it falls short of proving the positive semi-definiteness of the kernel.

In this paper, we address these issues in the following way:
We define a novel kernel for directed graphs based on [18] and the work of Chung [26] on directed Laplacians;We extend the work of [18] by incorporating additional node-level topological information using two well-known structural signatures, the Heat Kernel Signature [27] and the Wave Kernel Signature [28];We give a formal proof of the positive definiteness of the undirected kernel for the case where the Hamiltonian is the graph Laplacian and the starting state satisfies several constraints;We propose a simple yet efficient quantum algorithm to compute the kernel;We perform an empirical comparison of the performance of the kernel for both directed and undirected graphs and for different choices of the Hamiltonian.

We perform an extensive set of experimental evaluations and we find that:Adding the edge directionality information allows to better discriminate between the different classes, even when compared with other commonly used kernels for directed graphs;In most cases the incorporation of the node-level topological information results in a significant improvement over the performance of the original kernel;The optimal Hamiltonian (in terms of classification accuracy) depends on the dataset, as already suggested in [18];The constraints we enforce to ensure the positive definiteness of the kernel disrupt the phase of the initial state, leading to a decrease in classification accuracy.

The remainder of this paper is organised as follows. Section 2 reviews the fundamental concepts of graph theory and quantum mechanics needed to understand the present paper. Section 3 introduces the quantum-walk based similarity measure, analyses it from a theoretical perspectives, and illustrates how it can be adapted to work on both undirected and directed graphs. Finally, Section 4 discusses the results of our experimental evaluation and Section 5 concludes the paper.

## 2. Graphs and Quantum Walks

In this section we introduce several fundamental graph-theoretical and quantum mechanical notions. In particular, we show how to associate a density matrix to a graph using continuous-time quantum walks. The density matrix representation of the graphs will be then used in Section 3 to define the kernel for both directed and undirected graphs.

### 2.1. Elementary Graph-Theoretic Concepts

Let G(V,E) denote a directed graph, where *V* is a set of *n* vertices and E⊆V×V is a set of edges such that each e=(u,v)∈E has a start-vertex u∈V and an end-vertex v∈V. Please note that we sometimes omit (V,E) and simply write *G* when the vertex and edge sets are implicitely defined. The adjacency matrix *A* of *G* has elements
(1)$$A_{uv}=\left\{\begin{array}{c} 1\;\mathrm{if}\;(u,v)\in E\\ 0\;\mathrm{otherwise} \end{array}\right.$$

If $A_{uv}=A_{vu}=1$ we say that the edge e=(u,v) is bidirectional. If every edge of a graph *G* is bidirectional, i.e., the associated adjacency matrix *A* is symmetric, we say that *G* is an undirected graph.

For a node *u*, the degree of *u* is the number of nodes connected to *u* by an edge. For undirected graphs this is $d_{u}=\sum_{v\in V}A\left(u,v\right)$, while for directed graphs we distinguish between the in-degree $d_{u}^{in}=\sum_{v\in V}A\left(v,u\right)$ and the out-degree $d_{u}^{out}=\sum_{v\in V}A\left(u,v\right)$, which can be added up to give the total degree $d_{u}^{tot}=d_{u}^{in}+d_{u}^{out}$.

For undirected graphs, we can define the degree matrix *D* as the matrix with the $d_{u}$s as the diagonal elements and zero elsewhere. The graph Laplacian of *G* is then defined as L=D−A, and it can be interpreted as a combinatorial analogue of the discrete Laplace-Beltrami operator [29]. Similarly, we can define the normalised Laplacian $L=D^{-1/2}LD^{1/2}$. For directed graphs, things are less straightforward. In Section 3.1 we illustrate how to define both Laplacian and normalised Laplacian on directed graphs based on the work of Chung [26].

### 2.2. Quantum Walks on Graphs

Using the Dirac notation to denote as $|u\rangle$ the basis state corresponding to the walk being at vertex u∈V, the state of a continuous-time quantum walk on *G* at time is
(2)$$|\psi_{t}\rangle =\sum_{u\in V}\alpha_{u}\left(t\right)|u\rangle ,$$
where the amplitude $\alpha_{u}\left(t\right)\in C$ and $|\psi_{t}\rangle \in C^{\left|V\right|}$ are both complex. Moreover, we have that $\alpha_{u}\left(t\right)\alpha_{u}^{*}\left(t\right)$ gives the probability that at time *t* the walker is at the vertex *u*, and thus $\sum_{u\in V}\alpha_{u}\left(t\right)\alpha_{u}^{*}\left(t\right)=1$ and $\alpha_{u}\left(t\right)\alpha_{u}^{*}\left(t\right)\in \left[0,1\right]$, for all u∈V, $t\in R^{+}$.

The state of the walk evolves through time according to the Schrödinger equation
(3)$$\frac{\partial }{\partial t}|\psi_{t}\rangle =-iH|\psi_{t}\rangle \;$$
where H denotes the time-independent Hamiltonian. Given an initial state $|\psi_{0}\rangle$, the solution of the previous equation for time *t* is then
(4)$$|\psi_{t}\rangle =e^{-iHt}|\psi_{0}\rangle \;.$$

In the case of undirected graphs, the Laplacian matrix is usually chosen as the system Hamiltonian, i.e., H=L. However in theory any Hermitian operator encoding the graph structure can be used instead, e.g., the adjacency matrix or the normalised Laplacian matrix. Directed graphs pose an additional challenge, as the asymmetricity of the adjacency matrix implies that it cannot be used as the system Hamiltonian. However it is possible to define symmetric Laplacian and normalised Laplacian matrix representations for directed graphs, as explained in Section 3.1.

Equation (Equation 4) can be also rewritten in terms of the spectral decomposition of the Hamiltonian $H=\Phi \Lambda \Phi^{⊤}$, where Φ is the |V|×|V| matrix with the ordered eigenvectors of H as columns and Λ is the |V|×|V| diagonal matrix with the ordered eigenvalues $\lambda_{j}$ of H as elements. Since $\exp\left[-iHt\right]=\Phi \exp\left[-i\Lambda t\right]\Phi^{⊤}$, we can write the solution of the Schrödinger equation at time *t* as
(5)$$|\psi_{t}\rangle =\Phi e^{-i\Lambda t}\Phi^{⊤}|\psi_{0}\rangle \;.$$


### 2.3. From Quantum Walks to Graph Density Matrices

The density matrix representation is introduced in quantum mechanics as a way describe a system that is a statistical ensemble of pure states $|\psi_{i}\rangle$, each with probability $p_{i}$, i.e.,
(6)$$\rho =\sum_{i}p_{i}|\psi_{i}\rangle \langle \psi_{i}|\;.$$


Please note that density matrices are positive unit trace matrices. Given a density matrix ρ with eigenvalues $\lambda_{1},\ldots ,\lambda_{n}$, we can compute its von Neumann entropy [30] S(ρ) as
(7)$$S\left(\rho \right)=-\mathrm{tr}\left(\rho \log\rho \right)=-\sum_{i}\lambda_{i}\ln\;\lambda_{i}.$$


Moreover, given two density matrices ρ and σ we can compute their quantum Jensen-Shannon divergence (QJSD) [23,24] as
(8)$$QJSD\left(\rho ,\sigma \right)=S\left(\frac{\rho +\sigma }{2}\right)-\frac{1}{2}S\left(\rho \right)-\frac{1}{2}S\left(\sigma \right).$$

The QJSD is always well defined, symmetric and positive definite [23,24]. It can also be shown that QJSD(ρ,σ) is bounded, i.e., 0≤QJSD(ρ,σ)≤1, with the latter equality attained only if ρ and σ have support on orthogonal subspaces [18].

The QJSD has been shown to have several interesting properties required for a good distinguishability measure between quantum states [23,24]. In pattern recognition, it has been used to define several graph kernels [17,18,31]. This in turn is achieved by associating quantum states to graphs and using the QJSD between the states as a proxy of the similarity between the graphs.

More specifically, given a graph *G* and a continuous-time quantum walk with starting state $|\psi_{0}\rangle$, the idea is to associate a quantum state to *G* as follows. First, we let the quantum walk evolve according to Equation (Equation 4) until a time *T* is reached. Then, we define the quantum state with density matrix representation
(9)$$\rho_{T}=\frac{1}{T}\int_{0}^{T}\;|\psi_{t}\rangle \langle \psi_{t}|\;dt\;.$$

This state represents a uniform ensemble of pure states $|\psi_{t}\rangle$ each corresponding to a different stage of the quantum walk evolution. Different versions of QJSD-based graph kernels are based on different definitions of this density matrix [17,18]. In this paper we focus on the kernel of Rossi et al. [18], which was shown to have several advantages compared to its alternatives. However, while the original work of Rossi et al. [18] focused only on undirected graph, in the next sections we show how it can be extended to directed graphs. We also study how the choice of the Hamiltonian influences the kernel performance in a graph classification task and we investigate if the addition of structural information at node level has any benefit. While Rossi et al. [18] were unable to prove the positive definiteness of the kernel, we are able to provide a proof under the assumption that the Hamiltonian is the graph Laplacian and the starting state satisfies certain conditions. Finally, we propose a fully quantum procedure to compute the kernel.

## 3. Graph Similarity from Quantum Walks

  Given two undirected graphs $G_{1}\left(V_{1},E_{1}\right)$ and $G_{2}\left(V_{2},E_{2}\right)$, we merge them into a larger structure by establishing a complete set of connections between the two node sets. More specifically, we construct a graph G=(V,E) where $V=V_{1}\cup V_{2}$, $E=E_{1}\cup E_{2}\cup E_{12}$, and $\left(u,v\right)\in E_{12}$ only if $u\in V_{1}$ and $v\in V_{2}$. Figure 1 shows an example pair of graphs and their union.

The intuition of Rossi et al. [18] was to define two mixed quantum states on this new graph describing the time-evolution of two quantum walks specifically designed to amplify constructive and destructive interference effects. To this end, we define two independent quantum walks with starting states
(10)$$|\psi_{0}^{-}\rangle =\frac{\sum_{u\in V_{1}}d_{u}|u\rangle -\sum_{v\in V_{2}}d_{v}|v\rangle }{C}\;|\psi_{0}^{+}\rangle =\frac{\sum_{u\in V}d_{u}|u\rangle }{C}\;.$$

We can then associate the following quantum states to the graph,
(11)$$\rho_{T}^{-}=\frac{1}{T}\int_{0}^{T}\;|\psi_{t}^{-}\rangle \langle \psi_{t}^{-}|\;dt\;\rho_{T}^{+}=\frac{1}{T}\int_{0}^{T}\;|\psi_{t}^{+}\rangle \langle \psi_{t}^{+}|\;dt.$$

Given these two states, we finally make use of the QJSD to compute the similarity between the input graphs in terms of the dissimilarity between the $\rho_{T}^{-}$ and $\rho_{T}^{+}$, i.e.,
(12)$$k_{T}\left(G_{1},G_{2}\right)=QJSD\left(\rho_{T}^{-},\rho_{T}^{+}\right)=S\left(\frac{\rho_{T}^{-}+\rho_{T}^{+}}{2}\right)-\frac{1}{2}\left(S\left(\rho_{T}^{-}\right)+S\left(\rho_{T}^{+}\right)\right)\;.$$

Since the quantum states $\rho_{T}^{-}$ and $\rho_{T}^{+}$ represent the evolution of two quantum walks that emphasize destructive and constructive interference, respectively, we expect that the more similar the input graphs are, the more dissimilar $\rho_{T}^{-}$ and $\rho_{T}^{+}$ are. In the extreme case where $G_{1}$ and $G_{2}$ and isomorphic, it can be shown that $k_{T}\left(G_{1},G_{2}\right)=1$ [18].

In [18], the authors suggest to consider the limit of $\rho_{T}^{+}$ for T→∞, which in turn allows to rewrite Equation (Equation 11) as
(13)$$\rho_{\infty }^{+}=\sum_{\lambda \in \tilde{\Lambda }\left(H\right)}P_{\lambda }\rho_{0}^{+}P_{\lambda }^{⊤}\;,$$
where $\tilde{\Lambda }\left(H\right)$ is the set of distinct eigenvalues of the Hamiltonian (H), i.e., the eigenvalues λ with multiplicity μ(λ)=1, and $P_{\lambda }=\sum_{k=1}^{\mu (\lambda )}\phi_{\lambda ,k}\phi_{\lambda ,k}^{⊤}$ is the projection operator on the subspace spanned by the μ(λ) eigenvectors $\phi_{\lambda ,k}$ associated with λ. Similarly, one can work out the limit of $\rho_{T}^{-}$ for T→∞. This in turn has the effect of making the computation of the kernel easier in addition to allowing us to drop the time parameter *T*, yielding
(14)$$k_{\infty }\left(G_{1},G_{2}\right)=QJSD\left(\rho_{\infty }^{-},\rho_{\infty }^{+}\right)=S\left(\frac{\rho_{\infty }^{-}+\rho_{\infty }^{+}}{2}\right)-\frac{1}{2}\left(S\left(\rho_{\infty }^{-}\right)+S\left(\rho_{\infty }^{+}\right)\right)\;.$$


### 3.1. Extension to Directed Graphs

The kernel described in the previous subsection can only be computed for undirected graphs. In order to cope with directed graphs, we propose using the directed graph Laplacian introduced by Chung [26]. We first merge the two directed input graphs $G_{1}\left(V_{1},E_{1}\right)$ and $G_{2}\left(V_{2},E_{2}\right)$ as done in the undirected case, i.e., we create a complete set of undirected edges connecting the nodes of $V_{1}$ to the nodes of $V_{2}$. Note, however, that the resulting merge graph is directed, as a consequence of the edges in $E_{1}$ and $E_{2}$ being directed (see Figure 2). Hence we define the starting states $|\psi_{0}^{-}\rangle$ and $|\psi_{0}^{+}\rangle$ similarly to Equation (Equation 10), but using the out degree instead:(15)$$|\psi_{0}^{-}\rangle =\frac{\sum_{u\in V_{1}}d_{u}^{out}|u\rangle -\sum_{v\in V_{2}}d_{v}^{out}|v\rangle }{C}\;|\psi_{0}^{+}\rangle =\frac{\sum_{u\in V}d_{u}^{out}|u\rangle }{C}\;.$$

To compute the Hamiltonian and thus $\rho_{T}^{-}$ and $\rho_{T}^{+}$, we proceed as follows. Let us define the classical random walk transition matrix *M* for the directed graph *G* as the matrix with elements $M_{uv}=A_{uv}/d_{u}^{out}$, where we use the convention that when the out-degree of *u* is zero, i.e., $d_{u}^{out}=0$, we let $M_{uv}=0$.

From the Perron-Frobenius theorem we know that for a strongly connected directed graph, i.e., a directed graph where there exists a path between every pair of vertices and there are no sinks, the transition matrix *M* has a unique non-negative left eigenvector π [32], i.e., πM=λπ, where λ is the eigenvalue associated with π. According to the Perron-Frobenius theorem, if *M* is aperiodic we also have that λ=1, and thus πM=π, implying that the random walk on the directed graph *G* with transition matrix *M* will converge to a stationary probability distribution.

Let Π be the diagonal matrix with the elements of π on the diagonal. Then Chung [26] defines the Laplacian matrix of the directed graph *G* as
(16)$$L=\Pi -\frac{1}{2}\left(\Pi \;M+M^{⊤}\Pi \right).$$

Similarly, the normalised Laplacian of *G* is defined as
(17)$$\tilde{L}=I-\frac{1}{2}\left(\Pi^{\frac{1}{2}}\;M\;\Pi^{-\frac{1}{2}}+\Pi^{-\frac{1}{2}}M^{⊤}\Pi^{\frac{1}{2}}\right),$$
where *I* denotes the identity matrix.

Please note that while the adjacency matrix of a directed graph is clearly not Hermitian, both the normalised Laplacian and the Laplacian defined in Equations (Equation 16) and (Equation 17) are symmetric and thus can used as the Hamiltonian governing the quantum walks evolution. An alternative to this would have been to symmetrize the graph edges effectively making the graph undirected, however as we will show in the experimental part this causes the loss of important structural information.

We would like to stress that the method of Chung [26] is not the only way to associate a (normalised) Laplacian matrix to a directed graph. We are aware of at least one different method proposed by Bauer [33]. However we decide to focus on the definition proposed by Chung [26] as this has been successfully applied to the analysis of directed graphs in pattern recognition and network science [34,35,36].

### 3.2. Integrating Local Topological Information

In [31] it was shown that for the case of undirected attributed graphs $G_{1}\left(V_{1},E_{1},f_{1}\right)$ and $G_{2}\left(V_{2},E_{2},f_{2}\right)$, where $f_{i}$ is a function assigning attributes to the nodes of the graph $G_{i}$, it is possible to effectively incorporate the information on the pairwise similarity between the attributes of the two graphs by allowing the adjacency matrix of the merged graph to be weighted. More specifically, for each $u\in V_{1}$ and $v\in V_{2}$, the authors proposed to label the edge (u,v) with a real value $\omega (f_{1}\left(u\right),f_{2}\left(v\right))$ representing the similarity between $f_{1}\left(u\right)$ and $f_{2}\left(v\right)$.

While in this paper we only deal with unattributed graphs, we propose using a similar method to incorporate additional node-level structural information into the kernel. More specifically, given an undirected graph G(V,E) and a node v∈V, we capture the structure of the graph from the perspective of *v* using two well known spectral signatures based on the graph Laplacian, the Heat Kernel Signature (HKS) [27] and the Wave Kernel Signature (WKS) [28]. The signatures simulate a heat diffusion and wave propagation process on the graph, respectively. The result is in both cases a series of vectors characterising the graph topology centered at each node. In graph matching, particularly in computer vision applications [27,28], this is used to match pairs of nodes with similar (in an Euclidean sense) vectorial representations under the assumption that these describe structurally similar nodes in the original graph.

Given two nodes u,v∈V, we compute their spectral signatures $f_{1}\left(u\right)$ and $f_{1}\left(v\right)$ using either HKS or WKS and we label the edge (u,v) with the real value
(18)$$\omega \left(f_{1}\left(u\right),f_{2}\left(v\right)\right)=\|f_{1}\left(u\right)-f_{2}\left(v\right){\|}_{2}.$$

Please note that the signatures are computed on the original graphs, not on the merged one. Then the adjacency matrix of the graph obtained by merging $G_{1}$ and $G_{2}$ becomes
(19)$$A_{uv}=\left\{\begin{array}{c} \omega (f_{1}\left(u\right),f_{2}\left(v\right))\;\mathrm{if}\;u,v\in V\;\mathrm{and}\;u\neq v\\ 0\;\mathrm{otherwise}. \end{array}\right.$$

Figure 3 shows an example of two graphs with 2-dimensional signatures and their merged graph. For ease of presentation, the weight of each edge is shown by varying its thickness, i.e., the thicker the edge the higher the weight and the more similar the signatures of the connected nodes. Here we decided to compute the similarity between each pair of nodes in V, effectively turning $G_{1}$ and $G_{2}$ into two weighted cliques, i.e., complete weighted graphs, where the weights on the edges encode the pairwise nodes similarity. While at first sight it may seem like we are discarding the original structure of $G_{1}$ and $G_{2}$, note that this is actually encoded in the node signatures and thus the edge weights. In Section 4 we run an extensive set of experiments to determine if integrating node-level structural information improves the ability of the kernel to distinguish between structurally similar graphs.

### 3.3. Kernel Properties

Consider a pair of graphs $G_{1}$ and $G_{2}$, with *n* and *m* nodes respectively. Here we consider the case where we take the graph Laplacian to be the Hamiltonian of the system and we assume that the input graphs are undirected. Let $L_{1}$ and $L_{2}$ be the Laplacian of $G_{1}$ and $G_{2}$ respectively. Then the Laplacian of the merged graph is
(20)$$L=\left[\begin{array}{cc} L_{1}+mI_{n} & -1\;\;1_{n}1\;\;1_{m}{}^{⊤}\\ -1\;\;1_{m}1\;\;1_{n}{}^{⊤} & L_{2}+nI_{m} \end{array}\right].$$

We can provide a full characterization of the eigensystem of *L*. Let $|v\rangle$ and $|w\rangle$ be eigenvectors of $L_{1}$ and $L_{2}$, respectively, with corresponding eigenvalues *v* and *w*. Further, assume that $|v\rangle$ and $|w\rangle$ are not Fiedler vectors, i.e., $\langle 1\;\;1_{n}|v\rangle =0$ and $\langle 1\;\;1_{m}|w\rangle =0$. Then $|v,0_{m}\rangle$ is an eigenvector of *L* with eigenvalue v+m and $|0_{m},w\rangle$ is an eigenvector of *L* with eigenvalue w+n. The remaining eigenvectors of *L* are $|1\;\;1_{m+n}\rangle$ of eigenvalue 0 and $|m1\;\;1_{n},-n1\;\;1_{m}\rangle$ of eigenvalue m+n. In the following, we will denote with $P_{\lambda }^{1}$ the orthogonal projector on the eigenspace of $L_{1}$ of eigenvalue λ, if λ is an eigenvalue of $L_{1}$, 0 otherwise. For the special case of the 0 eigenspace, we eliminate the constant eigenvector $|1\;\;1_{n}\rangle$. $P_{\lambda }^{2}$ is similarly defined for $L_{2}$.

Under this eigendecomposition we can write $\rho_{\infty }^{+}$ and $\rho_{\infty }^{-}$ as the following sums of rank-1 matrices:
(21)$$\begin{array}{ccc} \rho_{\infty }^{+} & = & \sum_{\lambda \in \left(\tilde{\Lambda }\left(L_{1}\right)+m\right)\cup \left(\tilde{\Lambda }\left(L_{2}\right)+n\right)}|P_{\lambda -m}^{1}|\xi_{0}\rangle ,P_{\lambda -n}^{2}|\chi_{0}\rangle \rangle \langle P_{\lambda -m}^{1}|\xi_{0}\rangle ,P_{\lambda -n}^{2}|\chi_{0}\rangle |+\\ & & \frac{{\left(\langle 1\;\;1_{n}|\xi_{0}\rangle +\langle 1\;\;1_{m}|\chi_{0}\rangle \right)}^{2}}{{\left(m+n\right)}^{2}}|1\;\;1_{m+n}\rangle \langle 1\;\;1_{m+n}|+\frac{{\left(\frac{1}{n}\langle 1\;\;1_{n}|\xi_{0}\rangle -\frac{1}{m}\langle 1\;\;1_{m}|\chi_{0}\rangle \right)}^{2}}{{\left(m+n\right)}^{2}}|m1\;\;1_{n},-n1\;\;1_{m}\rangle \langle m1\;\;1_{n},-n1\;\;1_{m}|\\ & = & \sum_{\mu \in \left(\tilde{\Lambda }\left(L_{1}\right)-n\right)\cup \left(\tilde{\Lambda }\left(L_{2}\right)-m\right)}|P_{\mu +n}^{1}|\xi_{0}\rangle ,P_{\mu +m}^{2}|\chi_{0}\rangle \rangle \langle P_{\mu +n}^{1}|\xi_{0}\rangle ,P_{\mu +m}^{2}|\chi_{0}\rangle |+\\ & & \frac{{\left(\langle 1\;\;1_{n}|\xi_{0}\rangle +\langle 1\;\;1_{m}|\chi_{0}\rangle \right)}^{2}}{{\left(m+n\right)}^{2}}|1\;\;1_{m+n}\rangle \langle 1\;\;1_{m+n}|+\frac{{\left(\frac{1}{n}\langle 1\;\;1_{n}|\xi_{0}\rangle -\frac{1}{m}\langle 1\;\;1_{m}|\chi_{0}\rangle \right)}^{2}}{{\left(m+n\right)}^{2}}|m1\;\;1_{n},-n1\;\;1_{m}\rangle \langle m1\;\;1_{n},-n1\;\;1_{m}|, \end{array}$$
with the eigenvalues of *L* being λ=μ+m+n. Similarly,
(22)$$\begin{array}{ccc} \rho_{\infty }^{-} & = & \sum_{\mu \in \left(\tilde{\Lambda }\left(L_{1}\right)-n\right)\cup \left(\tilde{\Lambda }\left(L_{2}\right)-m\right)}|P_{\mu +n}^{1}|\xi_{0}\rangle ,-P_{\mu +m}^{2}|\chi_{0}\rangle \rangle \langle P_{\mu +n}^{1}|\xi_{0}\rangle ,-P_{\mu +m}^{2}|\chi_{0}\rangle |+\\ & & \frac{{\left(\langle 1\;\;1_{n}|\xi_{0}\rangle -\langle 1\;\;1_{m}|\chi_{0}\rangle \right)}^{2}}{{\left(m+n\right)}^{2}}|1\;\;1_{m+n}\rangle \langle 1\;\;1_{m+n}|+\frac{{\left(\frac{1}{n}\langle 1\;\;1_{n}|\xi_{0}\rangle +\frac{1}{m}\langle 1\;\;1_{m}|\chi_{0}\rangle \right)}^{2}}{{\left(m+n\right)}^{2}}|m1\;\;1_{n},-n1\;\;1_{m}\rangle \langle m1\;\;1_{n},-n1\;\;1_{m}|. \end{array}$$

Please note that each matrix in the summation is a rank-1 matrix and the matrices are orthogonal to each other as a consequence of the projectors being orthogonal to each others. This in turn implies that the size of the spectra of $\rho_{\infty }^{+}$ and $\rho_{\infty }^{-}$ is the same and it is equal to the number of distinct eigenvalues in $\left(\tilde{\Lambda }\left(L_{1}\right)-n\right)\cup \left(\tilde{\Lambda }\left(L_{2}\right)-m\right)$. More precisely, for each $\mu \in \left(\tilde{\Lambda }\left(L_{1}\right)-n\right)\cup \left(\tilde{\Lambda }\left(L_{2}\right)-m\right)$, excluding those derived from the two Fiedler vectors of $L_{1}$ and $L_{2}$, there exists a non-zero eigenvalue of both $\rho_{\infty }^{+}$ and $\rho_{\infty }^{-}$ of the form
$$\langle \xi_{0}|P_{\mu +n}^{1}|\xi_{0}\rangle +\langle \chi_{0}|P_{\mu +m}^{2}|\chi_{0}\rangle \;.$$

In addition, we have two eigenvalues of the form
$$\frac{{\left(\langle 1\;\;1_{n}|\xi_{0}\rangle +\langle 1\;\;1_{m}|\chi_{0}\rangle \right)}^{2}}{m+n}\;\mathrm{and}\;mn\frac{{\left(\frac{1}{n}\langle 1\;\;1_{n}|\xi_{0}\rangle -\frac{1}{m}\langle 1\;\;1_{m}|\chi_{0}\rangle \right)}^{2}}{m+n}\;,$$
for $L_{1}$, and
$$\frac{{\left(\langle 1\;\;1_{n}|\xi_{0}\rangle -\langle 1\;\;1_{m}|\chi_{0}\rangle \right)}^{2}}{m+n}\;\mathrm{and}\;mn\frac{{\left(\frac{1}{n}\langle 1\;\;1_{n}|\xi_{0}\rangle +\frac{1}{m}\langle 1\;\;1_{m}|\chi_{0}\rangle \right)}^{2}}{m+n}\;,$$
for $L_{2}$.

In other words the two density matrices, despite having different eigenspaces, have identical spectra induced from the eigenspaces orthogonal to the Fiedler vector and differ only in the values due to the extremal eigenvalues of *L*.

Turning our attention to the arithmetic mean of the two density matrices in Equation (Equation 14), we have
(23)$$\begin{array}{ccc} \frac{\rho_{\infty }^{+}+\rho_{\infty }^{-}}{2} & = & \sum_{\mu \in (\tilde{\Lambda }\left(L_{1}\right)-n)}|P_{\mu +n}^{1}|\xi_{0}\rangle ,0_{m}\rangle \langle P_{\mu +n}^{1}|\xi_{0}\rangle ,0_{m}|+\\ & & \sum_{\mu \in (\tilde{\Lambda }\left(L_{2}\right)-m)}|0_{n},-P_{\mu +m}^{2}|\chi_{0}\rangle \rangle \langle 0_{n},-P_{\mu +m}^{2}|\chi_{0}\rangle |+\\ & & \frac{{\langle 1\;\;1_{n}|\xi_{0}\rangle }^{2}+{\langle 1\;\;1_{m}|\chi_{0}\rangle }^{2}}{{\left(m+n\right)}^{2}}|1\;\;1_{m+n}\rangle \langle 1\;\;1_{m+n}|+\frac{{\left(\frac{\langle 1\;\;1_{n}|\xi_{0}\rangle }{n}\right)}^{2}+{\left(\frac{\langle 1\;\;1_{m}|\chi_{0}\rangle }{m}\right)}^{2}}{{\left(m+n\right)}^{2}}|m1\;\;1_{n},-n1\;\;1_{m}\rangle \langle m1\;\;1_{n},-n1\;\;1_{m}|. \end{array}$$


Again, this is a summation of rank-1 matrices orthogonal to one-another, resulting in eigenvalues of the form
$$\langle \xi_{0}|P_{\mu +n}^{1}|\xi_{0}\rangle \;\mathrm{and}\;\langle \chi_{0}|P_{\mu +m}^{2}|\chi_{0}\rangle \;,$$
corresponding to the (distinct) eigenvalues of $L_{1}$ and $L_{2}$, respectively, excluding the eigenspace component induced by the Fiedler vectors, as well as the two eigenvector from the extremal eigenvalues of *L*
$$\frac{{\langle 1\;\;1_{n}|\xi_{0}\rangle }^{2}+{\langle 1\;\;1_{m}|\chi_{0}\rangle }^{2}}{m+n}\;\mathrm{and}\;\frac{\frac{m}{n}{\langle 1\;\;1_{n}|\xi_{0}\rangle }^{2}+\frac{n}{m}{\langle 1\;\;1_{m}|\chi_{0}\rangle }^{2}}{m+n}.$$

Let us define the function
(24)xlog(x)=−xlog(x).

We can now compute the QJSD kernel between the graphs $G_{1}$ and $G_{2}$ as
(25)$$\begin{array}{ccc} k_{\infty }\left(G_{1},G_{2}\right) & = & QJSD\left(\rho_{\infty }^{-},\rho_{\infty }^{+}\right)=S\left(\frac{\rho_{\infty }^{-}+\rho_{\infty }^{+}}{2}\right)-\frac{1}{2}\left(S\left(\rho_{\infty }^{-}\right)+S\left(\rho_{\infty }^{+}\right)\right)\\ & = & \sum_{\mu \in (\tilde{\Lambda }\left(L_{1}\right)-n)}\mathrm{xlog}\left(\langle \xi_{0}|P_{\mu +n}^{1}|\xi_{0}\rangle \right)+\sum_{\mu \in \left(\tilde{\Lambda }\left(L_{2}\right)-m\right)}\mathrm{xlog}\left(\langle \chi_{0}|P_{\mu +m}^{2}|\chi_{0}\rangle \right)\\ & & -\sum_{\mu \in \left(\tilde{\Lambda }\left(L_{1}\right)-n\right)\cup \left(\tilde{\Lambda }\left(L_{2}\right)-m\right)}\mathrm{xlog}\left(\langle \xi_{0}|P_{\mu +n}^{1}|\xi_{0}\rangle +\langle \chi_{0}|P_{\mu +m}^{2}|\chi_{0}\rangle \right)\\ & & +\mathrm{xlog}\left(\frac{{\langle 1\;\;1_{n}|\xi_{0}\rangle }^{2}+{\langle 1\;\;1_{m}|\chi_{0}\rangle }^{2}}{m+n}\right)+\mathrm{xlog}\left(\frac{\frac{m}{n}{\langle 1\;\;1_{n}|\xi_{0}\rangle }^{2}+\frac{n}{m}{\langle 1\;\;1_{m}|\chi_{0}\rangle }^{2}}{m+n}\right)\\ & & -\frac{1}{2}\mathrm{xlog}\left(\frac{{\left(\langle 1\;\;1_{n}|\xi_{0}\rangle +\langle 1\;\;1_{m}|\chi_{0}\rangle \right)}^{2}}{m+n}\right)-\frac{1}{2}\mathrm{xlog}\left(mn\frac{{\left(\frac{1}{n}\langle 1\;\;1_{n}|\xi_{0}\rangle -\frac{1}{m}\langle 1\;\;1_{m}|\chi_{0}\rangle \right)}^{2}}{m+n}\right)\\ & & -\frac{1}{2}\mathrm{xlog}\left(\frac{{\left(\langle 1\;\;1_{n}|\xi_{0}\rangle -\langle 1\;\;1_{m}|\chi_{0}\rangle \right)}^{2}}{m+n}\right)-\frac{1}{2}\mathrm{xlog}\left(mn\frac{{\left(\frac{1}{n}\langle 1\;\;1_{n}|\xi_{0}\rangle +\frac{1}{m}\langle 1\;\;1_{m}|\chi_{0}\rangle \right)}^{2}}{m+n}\right). \end{array}$$

Assuming that the initial states are normalised in such a way that $\langle 1\;\;1_{n}|\xi_{0}\rangle =\langle 1\;\;1_{m}|\chi_{0}\rangle =0$ and $\langle \xi_{0}|\xi_{0}\rangle =\langle \chi_{0}|\chi_{0}\rangle =\frac{1}{2}$, the last terms disappear and we have
(26)$$\begin{array}{ccc} k_{\infty }\left(G_{1},G_{2}\right) & = & \sum_{\mu \in (\tilde{\Lambda }\left(L_{1}\right)-n)}\mathrm{xlog}\left(\langle \xi_{0}|P_{\mu +n}^{1}|\xi_{0}\rangle \right)+\sum_{\mu \in \left(\tilde{\Lambda }\left(L_{2}\right)-m\right)}\mathrm{xlog}\left(\langle \chi_{0}|P_{\mu +m}^{2}|\chi_{0}\rangle \right)\\ & & -\sum_{\mu \in \left(\tilde{\Lambda }\left(L_{1}\right)-n\right)\cup \left(\tilde{\Lambda }\left(L_{2}\right)-m\right)}\mathrm{xlog}\left(\langle \xi_{0}|P_{\mu +n}^{1}|\xi_{0}\rangle +\langle \chi_{0}|P_{\mu +m}^{2}|\chi_{0}\rangle \right)\\ & = & \frac{1}{2}\sum_{\mu \in (\tilde{\Lambda }\left(L_{1}\right)-n)}\mathrm{xlog}\left(2\langle \xi_{0}|P_{\mu +n}^{1}|\xi_{0}\rangle \right)+\frac{1}{2}\log\left(2\right)+\frac{1}{2}\sum_{\mu \in \left(\tilde{\Lambda }\left(L_{2}\right)-m\right)}\mathrm{xlog}\left(2\langle \chi_{0}|P_{\mu +m}^{2}|\chi_{0}\rangle \right)\\ & & +\frac{1}{2}\log\left(2\right)-\sum_{\mu \in \left(\tilde{\Lambda }\left(L_{1}\right)-n\right)\cup \left(\tilde{\Lambda }\left(L_{2}\right)-m\right)}\mathrm{xlog}\left(\frac{2\langle \xi_{0}|P_{\mu +n}^{1}|\xi_{0}\rangle +2\langle \chi_{0}|P_{\mu +m}^{2}|\chi_{0}\rangle }{2}\right)\\ & = & \log\left(2\right)-\left[\sum_{\mu \in \left(\tilde{\Lambda }\left(L_{1}\right)-n\right)\cup \left(\tilde{\Lambda }\left(L_{2}\right)-m\right)}\mathrm{xlog}\left(\frac{2\langle \xi_{0}|P_{\mu +n}^{1}|\xi_{0}\rangle +2\langle \chi_{0}|P_{\mu +m}^{2}|\chi_{0}\rangle }{2}\right)\right.\\ & & \left.-\frac{1}{2}\sum_{\mu \in (\tilde{\Lambda }\left(L_{1}\right)-n)}\mathrm{xlog}\left(2\langle \xi_{0}|P_{\mu +n}^{1}|\xi_{0}\rangle \right)-\frac{1}{2}\sum_{\mu \in \left(\tilde{\Lambda }\left(L_{2}\right)-m\right)}\mathrm{xlog}\left(2\langle \chi_{0}|P_{\mu +m}^{2}|\chi_{0}\rangle \right)\right]\;. \end{array}$$

This leads us to the following final observation. Let $f_{1}$ and $f_{2}$ be two discrete distributions with values in $\tilde{\Lambda }\left(L_{1}\right)-n$ and $\tilde{\Lambda }\left(L_{2}\right)-m$, respectively, defined as
(27)$$\begin{array}{ccc} f_{1}\left(x\right) & = & \sum_{\mu \in (\tilde{\Lambda }\left(L_{1}\right)-n)}\left(2\langle \xi_{0}|P_{\mu +n}^{1}|\xi_{0}\rangle \right)\delta \left(x-\mu \right) \end{array}$$
(28)$$\begin{array}{ccc} f_{2}\left(x\right) & = & \sum_{\mu \in \left(\tilde{\Lambda }\left(L_{2}\right)-m\right)}\left(2\langle \chi_{0}|P_{\mu +m}^{2}|\chi_{0}\rangle \right)\delta \left(x-\mu \right)\;, \end{array}$$
where δ(x) is the Dirac delta function. Then we have
(29)$$k_{\infty }\left(G_{1},G_{2}\right)=QJSD\left(\rho_{\infty }^{-},\rho_{\infty }^{+}\right)=1-CJSD\left(f_{1},f_{2}\right)\;,$$
where CJSD denotes the classical Jensen-Shannon divergence. Now we can prove the following:

**Theorem** **1.**
*Using the Laplacian as the Hamiltonian and normalizing $|\xi_{0}\rangle$ and $|\chi_{0}\rangle$ such that $\langle \xi_{0}|\xi_{0}\rangle =\langle \xi_{0}|\xi_{0}\rangle =\frac{1}{2}$ and $\langle 1\;\;1_{n}|\xi_{0}\rangle =\langle 1\;\;1_{m}|\xi_{0}\rangle =0$, then the quantum Jensen-Shannon kernel $k_{\infty }\left(G_{1},G_{2}\right)=QJSD\left(\rho_{\infty }^{-},\rho_{\infty }^{+}\right)$ is positive definite.*


**Proof.** The proof descends directly from the derivations above and from the fact that the classical Jensen-Shannon divergence is conditionally negative-definite [37]. □

### 3.4. Kernel Computation

The issue of efficiently computing the QJSD kernel is that of computing the entropies of $\rho_{\infty }^{-}$, $\rho_{\infty }^{+}$, and of their mean. Here we will discuss a general quantum approach to estimate the entropy of an infinite-time average mixing matrix starting from a pure state. Following [18], we write the infinite-time average mixing matrix as
(30)$$\rho_{\infty }=\sum_{\lambda \in \tilde{\Lambda }\left(H\right)}P_{\lambda }\rho_{0}P_{\lambda }\;,$$
where H is the Hamiltonian and $P_{\lambda }$ is the projector onto its λ eigenspace. If $\rho_{0}=|\phi \rangle \langle \phi |$ is a pure state, then the eigenvectors corresponding to non-zero eigenvalues of $\rho_{\infty }$ are of the form
(31)$$P_{\lambda }|\phi \rangle \;,$$
for $\lambda \in \tilde{\Lambda }\left(H\right)$, and the corresponding eigenvalues are
(32)$$\langle \phi |P_{\lambda }|\phi \rangle \;.$$

We already used this property in the previous section for the special case of the graph Laplacian as the Hamiltonian, but it holds in general.

It is important to note that the lambdas here are energy levels of the system, i.e., possible observed values using the Hamiltonian as an observable, and the probability of observing an energy state λ from the state $\rho_{0}=|\phi \rangle \langle \phi |$ is $\langle \phi |P_{\lambda }|\phi \rangle$. This means that the von Neumann entropy of $\rho_{\infty }$ is equivalent to the classic Shannon entropy of the energy levels of the (pure) initial state, i.e., the entropy of the possible observations from the pure initial state using the Hamiltonian as the observable.

The quantum algorithm to estimate the von Neumann entropy of the infinite-time average mixing matrix $\rho_{\infty }=|\phi \rangle \langle \phi |$ is as follows. First, we create several systems in the initial pure state $|\phi \rangle$ and we observe their energy using H as the observable. Then we use any entropy estimator for finite (discrete) distributions, using the energies as samples. Algorithms such as [38] or [39] are particularly interesting because they work well in the under-sampled case.

In the special case considered in Theorem 1, we can make the estimation even more efficient noting that we can observe independently from $L_{1}-nI_{n}$ and $L_{2}-mI_{m}$, and in particular we can perform a single set of observation for each graph and reuse them in every pairwise computation for the kernel. In this case, the samples from the first graph are used to estimate the entropy $S(f_{1})$, the samples from the second graph are used to estimate $S(f_{2})$, and the two sample-sets are merged to estimate $S\left(\frac{f_{1}+f_{2}}{2}\right)$.

## 4. Experiments

In this section, we evaluate the accuracy of the proposed kernel in several classification tasks. Our aim is to investigate (1) the importance of the choice of the Hamiltonian; (2) the integration of node-level structural signatures; and (3) the ability to incorporate edge directionality information using the Laplacians proposed by Chung [26]. To this end, we make use of the following datasets of undirected graphs (see Table 1):

**MUTAG** [40] is a dataset consisting originally of 230 chemical compounds tested for mutagenicity in Salmonella typhimurium [41]. Among the 230 compounds, however, only 188 (125 positive, 63 negative) are considered to be learnable and thus are used in our simulations. The 188 chemical compounds are represented by graphs. The aim is predicting whether each compound possesses mutagenicity.

**PPIs** (Protein-Protein Interaction) is a dataset collecting protein-protein interaction networks related to histidine kinase [42] (40 PPIs from Acidovorax avenae and 46 PPIs from Acidobacteria) [43]. The graphs describe the interaction relationships between histidine kinase in different species of bacteria. Histidine kinase is a key protein in the development of signal transduction. If two proteins have direct (physical) or indirect (functional) association, they are connected by an edge. The original dataset comprises 219 PPIs from 5 different kinds of bacteria with the following evolution order (from older to more recent): Aquifex 4 and Thermotoga 4 PPIs from Aquifex aelicus and Thermotoga maritima, Gram-Positive 52 PPIs from Staphylococcus aureus, Cyanobacteria 73 PPIs from Anabaena variabilis and Proteobacteria 40 PPIs from Acidovorax avenae. There is an additional class (Acidobacteria 46 PPIs) which is more controversial in terms of the bacterial evolution since they were discovered.

**PTC** (Predictive Toxicology Challenge) dataset records the carcinogenicity of several hundred chemical compounds for Male Rats (MR), Female Rats (FR), Male Mice (MM) and Female Mice (FM) [44]. These graphs are very small and sparse. We select the graphs of Male Rats (MR) for evaluation. There are 344 test graphs in the MR class.

**COIL** Columbia Object Image Library consists of 3D objects images of 100 objects [45]. There are 72 images per object taken in order to obtain 72 views from equally spaced viewing directions. For each view a graph was built by triangulating the extracted Harris corner points. In our experiments, we use the gray-scale images of five objects.

**NCI1** The anti-cancer activity prediction dataset consists of undirected graphs representing chemical compounds screened for activity against non-small cell lung cancer lines [46]. Here we use only the connected graphs in the dataset (3530 out of 4110).

We also make use of the following directed graphs datasets:

**Shock** The Shock dataset consists of graphs from a database of 2D shapes [47]. Each graph is a medial axis-based representation of the differential structure of the boundary of a 2D shape. There are 150 graphs divided into 10 classes, each containing 15 graphs. The original version contains directed trees each with a root node, the undirected version has been created by removing the directionality.

**Alzheimer** The dataset is obtained from the Alzheimer’s Disease Neuroimaging Initiative (ADNI) [48] and concerns interregional connectivity structure for functional magnetic resonance imaging (fMRI) activation networks for normal and Alzheimer subjects. Each image volume is acquired every two seconds with blood oxygenation level dependent signals (BOLD). The fMRI voxels here have been aggregated into larger regions of interest (ROIs). The different ROIs correspond to different anatomical regions of the brain and are assigned anatomical labels to distinguish them. There are 96 anatomical regions in each fMRI image. The correlation between the average time series in different ROIs represents the degree of functional connectivity between regions which are driven by neural activities [49]. Subjects fall into four categories according to their degree of disease severity: AD—full Alzheimer’s (30 subjects), LMCI—Late Mild Cognitive Impairment (34 subjects), EMCI—Early Mild Cognitive Impairment (47 subjects), HC—Normal Healthy Controls (38 subjects). The LMCI subjects are more severely affected and close to full Alzheimerś, while the EMCI subjects are closer to the healthy control group (Normal). A directed graph with 96 nodes is constructed for each patient based on the magnitude of the correlation and the sign of the time-lag between the time-series for different anatomical regions. To model causal interaction among ROIs, the directed graph uses the time lagged cross-correlation coefficients for the average time series for pairs of ROIs. We detect directed edges by finding the time-lag that results in the maximum value of the cross-correlation coefficient. The direction of the edge depends on whether the time lag is positive or negative. We then apply a threshold to the maximum values to retain directed edges with the top 40% of correlation coefficients. This yields a binary directed adjacency matrix for each subject, where the diagonal elements are set to zero. Those ROIs which have missing time series data are discarded. In order to fairly evaluate the influence caused by edges directionality, an undirected copy has been created as well. In particular, let $A_{d}$ be the adjacency matrix of a directed graph. Then its projection over the symmetric matrices space will be given by $(A_{d}+A_{d}^{⊤})/2$.

### 4.1. Experimental Framework

With these datasets to hand, we aim to solve a graph classification task using as binary C-Support Vector Machine (C-SVM) [19]. We perform 10-fold cross validation, where for each sample we independently tune the value of *C*, the SVM regularizer constant, by considering the training data from that sample. The process is averaged over 100 random partitions of the data, and the results are reported in terms of average accuracy ± standard error. Recall that in classification tasks, the accuracy is intended as the fraction of the data occurrences that are assigned the correct class label. In our case, an occurrence is a graph. Moreover, we contrast the performance of the kernel with that of other well-established alternative graph kernels, namely the shortest-path kernel [15] (SP), the classic random walk kernel [9] (RW) and the the Weisfeiler-Lehman subtree kernel [16] (WL).

We divide our experiments into two parts. We first evaluate the influence of the Hamiltonian on the classification accuracy, where the Hamiltonian is chosen to be one between (1) the adjacency matrix ($QJSD_{A}$); (2) the Laplacian ($QJSD_{L}$); or (3) the normalised Laplacian of the graph ($QJSD_{NL}$). Here we also consider the positive definite version of the kernel discussed in Section 3.3 ($QJSD_{*}$). In the same experiment, we test if the addition of node signatures benefits the kernel performance, where the signature is either the (1) HKS or the (2) WKS, denoted as ${}^{hk}$ and ${}^{wk}$ respectively. In the second experiment we test the performance of our kernel for the classification of directed graphs. Please note that all the alternative kernels we consider are also able to cope with directed graphs. Here we test both our competitiveness against these kernels as well as the increase or decrease in performance compared to the case where we remove the edge directionality information.

### 4.2. Undirected Graphs

Table 2 shows the results of the classification experiments on undirected graphs. The best and second best performing kernels are highlighted in bold and italic, respectively. While the kernels based on the quantum Jensen-Shannon divergence yield the highest accuracy in 4 out of 5 datasets, it is clear that no specific combination of Hamiltonian and structural signatures is optimal in all cases. However, we observe that with the exception of the PPI dataset, both the choice of the Hamiltonian and the use or not of structural signatures has little or no impact on the performance of the kernel, which remains highly competitive when compared to the shortest-path, random walks, graphlet, and Weisfeiler-Lehman kernels. The performance gain is particularly evident when compared to the kernel based on classical random walks, which is consistently outperformed by the kernels based on quantum walks. This in turn is a confirmation of the increased ability of capturing subtle structural similarities of quantum walks with respect to their classical counterpart, as well as a clear statement of the robustness of the proposed kernel to the choice of its parameters. We do observe a tendency of the Laplacian + WKS combination to perform better on datasets with average graph size >100 (PPI and COIL), however this would need more extensive experiments to be validated.

We would like to stress that there are indeed alternative strategies to the one proposed in Section 3.2 to integrate the node signatures information. For example, instead of creating a pair of weighted cliques, one could retain the original graphs structure and compute the similarity only between nodes connected by edges in $E_{1}$, $E_{2}$, and $E_{12}$. Another approach could have been to restrict the weighting only to the edges in $E_{12}$, similarly to what done by Rossi et al. in [31]. However we found little or no difference in terms of classification accuracy between these alternative solutions and thus we decided to omit these results for ease of presentation.

Finally, the performance of the positive definite kernel $QJSD_{*}$ deserves a closer inspection. In contrast to the other quantum-walk based kernels, in this case the performance is often significantly lower and the dependency on the addition of the node-level structural signatures higher. Unlike the other kernels, in this case we had to enforce several constraints on the initial state in order to guarantee the positive definiteness of the resulting kernel (see Section 3.3). However one of this constraints, i.e., the requirement that the two parts of the initial amplitude vector corresponding to $G_{1}$ and $G_{2}$ both sum to zero, causes the phase of the initial state to differ from that of the other kernels. While the initial states defined in Equation (Equation 10) are such that $|\psi_{0}^{+}\rangle$ has equal phase on both $G_{1}$ and $G_{2}$, while $|\psi_{0}^{-}\rangle$ has opposite phase on $G_{1}$ and $G_{2}$, enforcing the constraint of Section 3.3 leads to a change of sign (and thus of phase) for the smallest (in absolute value) components of the initial state. This in turn invalidates the assumption that $\rho_{\infty }^{+}$ and $\rho_{\infty }^{-}$ are designed to stress constructive and destructive interference, respectively, and thus affects the ability of the kernel to distinguish between similar/dissimilar graphs.

### 4.3. Directed Graphs

Table 3 shows the results of the classification experiments on directed graphs. We compare the kernels on the two directed graphs dataset, as well as their undirected versions, and we highlight again the best performing kernel for each dataset. Please note that when the graphs are directed we cannot choose the (potentially asymmetric) adjacency matrix as the Hamiltonian. We first observe that preserving the edge directionality information leads to a higher or similar classification accuracy for most kernels, with the exception of the shortest-path kernel on the Shock dataset. Please note that the graphs in this dataset are trees, and as such in the directed version there may be pairs of nodes for which a shortest-path does not exist, leading to the observed decrease in classification performance. In general, we observe that both in the directed and undirected case our kernel is the best performing, with the normalised Laplacian being the Hamiltonian that leads to the highest classification accuracy in 3 out of 4 datasets.

## 5. Conclusions

We studied the use of continuous-time quantum walks and the quantum Jensen-Shannon divergence for the purpose of measuring the similarity between two graphs. We performed a thorough analysis of the similarity measure originally introduced in [18], showing that: (1) the kernel can be extended to deal with directed graphs, allowing to better discriminate between the different classes, even when compared with other commonly used kernels for directed graphs; (2) while it is possible to incorporate additional node-level topological information in the kernel, this results in a significant improvement over the performance of the original kernel; (3) the optimal Hamiltonian (in terms of classification accuracy) depends on the dataset; (4) when appropriate conditions are met, the kernel is proved to be positive-definite, however (5) its performance suffers from a loss of structure in the design of the initial state phase; (6) it is possible to efficiently compute the kernel using a fully quantum procedure.

Despite having made important progress toward a better understanding of the quantum Jensen-Shannon divergence kernel, several questions remain unanswered. As such, future work should aim at understanding the role played by the initial state in the performance of the kernel, as well as considering alternative divergence measures and different ways of encoding the edge directionality information. Another idea could be to investigate if we can link the similarity between two graphs to the quantumness [50] of a quantum walk on their merged graph, based on the observation that isomorphic graphs result in a highly symmetric merged graph where strong interference effects can lead the quantum walk to behave significantly differently from its classical counterpart.

## Figures and Tables

**Figure 1 entropy-21-00328-f001:**
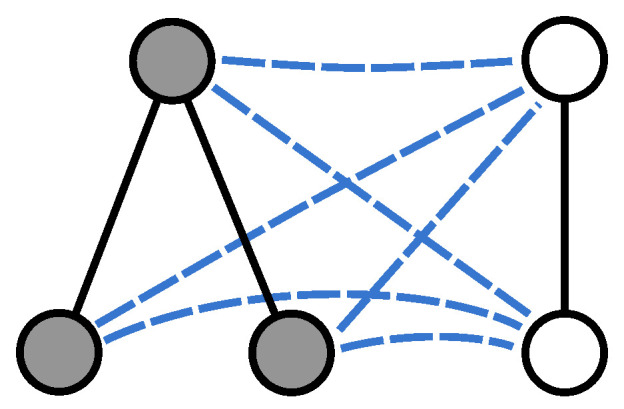
Given two graphs $G_{1}\left(V_{1},E_{1}\right)$ and $G_{2}\left(V_{2},E_{2}\right)$ we build a new graph G=(V,E) where $V=V_{1}\cup V_{2}$, $E=E_{1}\cup E_{2}$ and we add a new edge (u,v) between each pair of nodes $u\in V_{1}$ and $v\in V_{2}$.

**Figure 2 entropy-21-00328-f002:**
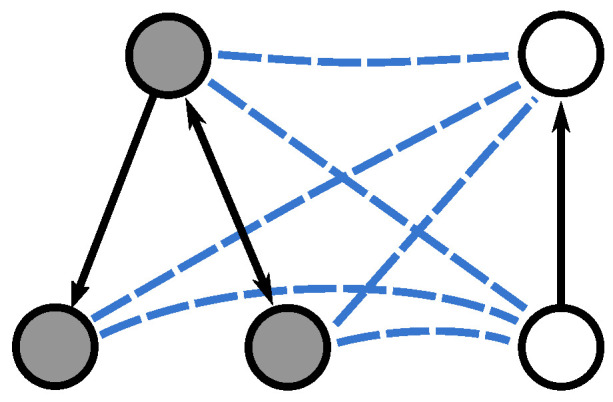
The graph obtained by merging two directed graphs.

**Figure 3 entropy-21-00328-f003:**
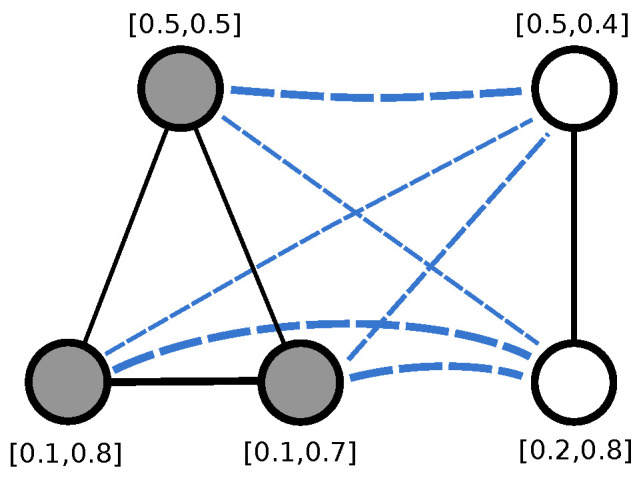
The graph obtained by merging two undirected graphs with 2-dimensional node signatures. The tickness of the edges is proportional to the similarity between the signatures of the nodes being connected.

**Table 1 entropy-21-00328-t001:** Information on the graph datasets.

Datasets	MUTAG	PPI	PTC	COIL	NCI1	SHOCK	ALZ
Max # vertices	28	232	109	241	106	33	96
Min # vertices	10	3	2	72	3	4	96
Avg # vertices	17.93	109.60	25.56	144.97	29.27	13.16	96
# graphs	188	86	344	360	3530	150	149
# classes	2	2	2	5	2	10	4

**Table 2 entropy-21-00328-t002:** Classification accuracy (±standard error) on undirected graph datasets. The best and second best performing kernels are highlighted in bold and italic, respectively.

Kernel	MUTAG	PPI	PTC	NCI1	COIL
QJSD${}_{A}$	86.72±0.14	78.71±0.30	56.09±0.15	66.90±0.03	69.90±0.08
QJSD${}_{L}$	84.92±0.18	73.79±0.42	59.70±0.16	69.48±0.03	70.72±0.07
QJSD${}_{NL}$	87.10±0.14	74.63±0.37	55.16±0.18	66.36±0.03	69.61±0.10
QJSD${}_{A}^{hk}$	88.51±0.13	81.56±0.34	58.76±0.14	63.88±0.03	69.68±0.06
QJSD${}_{L}^{hk}$	86.36±0.16	77.08±0.28	57.63±0.13	64.96±0.02	70.24±0.06
QJSD${}_{NL}^{hk}$	87.79±0.12	74.38±0.37	58.45±0.16	64.01±0.04	70.55±0.09
QJSD${}_{A}^{wk}$	85.97±0.14	74.91±0.32	58.91±0.13	63.52±0.05	70.48±0.06
QJSD${}_{L}^{wk}$	85.81±0.16	84.66±0.26	58.01±0.13	64.45±0.03	71.34±0.05
QJSD${}_{NL}^{wk}$	87.61±0.16	74.74±0.30	57.46±0.16	63.34±0.04	70.31±0.06
QJSD${}_{*}$	75.47±0.18	56.50±0.41	58.15±0.12	62.89±0.02	13.51±0.16
QJSD${}_{*}^{hk}$	70.99±0.12	70.53±0.31	55.29±0.13	67.19±0.01	34.24±0.22
QJSD${}_{*}^{wk}$	79.44±0.16	70.21±0.40	57.65±0.14	64.20±0.02	58.63±0.12
SP	84.98±0.16	66.40±0.31	56.89±0.71	65.44±0.04	70.50±0.13
RW	78.02±0.20	69.94±0.27	55.59±0.01	58.80±0.04	21.03±0.22
GR	81.93±0.17	52.34±0.42	56.20±0.10	62.28±0.02	67.22±0.11
WL	84.62±0.23	79.93±0.35	55.64±0.20	78.55±0.04	31.33±0.21

**Table 3 entropy-21-00328-t003:** Classification accuracy (±standard error) on directed graph datasets, where ${}_{d}$ and ${}_{u}$ denote the directed and undirected versions of the datasets, respectively. The best and second best performing kernels are highlighted in bold and italic, respectively.

Kernel	ALZ${}_{d}$	ALZ${}_{u}$	SHOCK${}_{d}$	SHOCK${}_{u}$
QJSD${}_{A}$	-	65.87±0.25	-	41.48±0.15
QJSD${}_{L}$	79.26±0.24	60.42±0.23	45.89±0.23	35.77±0.21
QJSD${}_{NL}$	82.07±0.17	61.45±0.22	46.05±0.20	44.38±0.21
SP	59.86±0.25	58.00±0.29	22.09±0.29	40.16±0.24
RW	79.06±0.21	60.75±0.25	30.45±0.26	24.34±0.28
GR	79.00±0.20	64.34±0.27	28.91±0.30	29.39±0.28
WL	70.87±0.27	59.46±0.35	38.74±0.27	35.78±0.26

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
