# Peer review of "Can a Quantum Walk Tell Which Is Which?A Study of Quantum Walk-Based Graph Similarity"

_entropy, 2019, doi:10.3390/e21030328_

Reviewer 1 Report

This paper is about the problem of measuring the similarity between 2 graphs using CTQW and comparing their time evolution. Authors consider both cases: directed and undirected graph.

The paper is well written, it is easy to read. However, Table 3 is the most important part of the paper and I think authors can explain it better. For example, why RW has a poor classification accuracy? or SP?

Authors propose to study of quantum walk based graph similarity. They claim that their measure can effectively incorporate new information of graph (direction). Also, they propose a novel kernel, extend their own work (2015). The research propose new algorithms in this topic.

There are many problems modeled as directed graph.

They made several experiments and their results show that is possible to measure similarity between graphs.

In general I consider this paper is clear.

I suggest to improve the reference, the most recent is since 2015 and I suggest to include this

How Quantum is a "Quantum Walk"?

F. Shahbeigi, S. J. Akhtarshenas, A. T. Rezakhani, arXiv:1802.07027

Author Response

Following the reviewer’s suggestion, we’ve added the reference mentioned in the review noting that this or other quantumness measures could perhaps be explored in conjunction with our framework (or a modified version of it) for the purpose of measuring graph similarity. The added sentence can be found at the end of the Conclusion section.

We have also added a possible explanation of the decrease in performance for the shortest-path kernel observed in table 3. First, we apologise for mistakenly reporting the wrong accuracy for the random walk kernel when editing table 3. The table has now been updated. Our kernel remains the best performing one, but indeed the accuracy of the RW kernel increases when the edge directionality is taken into account. The only kernel that shows a marked decrease in the performance when going from undirected to directed graphs is the shortest path kernel (SP). We added the following sentence to section 4.3 to account for this behaviour: “Note that the graphs in this dataset are trees, and as such in the directed version there may be pairs of nodes for which a shortest-path does not exist, leading to the observed decrease in classification performance.”

Reviewer 2 Report

The authors analyze the possibility to use continuous time quantum walk for distinguishing underlying graphs. In particular, they merge the two graphs by a particular procedure and consider a time evolution on this new graph with two distinct initial state which are defined in such a way that they amplify interference effects between the nodes of the two original graphs. Then they define time averaged density matrices arising from the two initial states and evaluate their quantum Jensen-Shannon divergence, which serves as the measure of similarity of the original graphs. They focus on the asymptotic limit, where the time averaged density matrices can be calculated with the help of the spectral decomposition of the quantum walk Hamiltonian.

This approach to testing graph similarity was proposed in an earlier publication of two of the authors. In the present paper, they extend the ideas to directed graphs using the directed Laplacian as the Hamiltonian of the quantum walk. In addition, they prove that under certain conditions the similarity measure is positive definite and discuss how the measure can be estimated by measuring the energy and estimating the Shannon entropy of the observations of different energy levels. The proposed similarity measure is tested on several directed and undirected graph datasets with good results.

Overall the paper is well written, all concepts are introduced in a comprehensible way. The results are interesting and deserves a publication in the journal Entropy. 

I have noticed one typo which should be corrected - on page 1, the author of reference [7] is Childs (not Child). Apart from this minor misprint the paper can be published in its present form.

Author Response

Many thanks for the encouraging review. We have corrected the typo in the citation of [7].